# Downregulation of Aging-Associated Gene SUCLG1 Marks the Aggressiveness of Liver Disease

**DOI:** 10.3390/cancers17030339

**Published:** 2025-01-21

**Authors:** Desislava K. Tsoneva, Alessandro Napoli, Mariya Teneva, Tommaso Mazza, Manlio Vinciguerra

**Affiliations:** 1Department of Medical Genetics, Medical University of Varna, 9002 Varna, Bulgaria; 2Department of Stem Cell Biology and Transplantology, Research Institute of the Medical University of Varna, 9002 Varna, Bulgaria; 3Bioinformatics Laboratory, Fondazione IRCCS Casa Sollievo della Sofferenza, 71013 S. Giovanni Rotondo, FG, Italy; 4Faculty of Science, Liverpool John Moores University, Liverpool L3 3AF, UK

**Keywords:** cancer, liver disease, gene expression, aging

## Abstract

Benign conditions such as nonalcoholic fatty liver disease (NAFLD) can progress to severe and deadly malignant liver diseases such as hepatocellular carcinoma (HCC) and cholangiocarcinoma (CC). Aging is a major factor contributing to the progression of such diseases through mechanisms such as chronic inflammation, immune system decline, and genetic alterations. However, little is known about the molecular alterations shared between aging and the acquired pathological changes with the advance of liver disease. In this study, we investigated how age-related gene expression changes associated with the progression of liver disease. By analyzing tissue samples from various liver disease stages, we found decreased levels of the gene SUCLG1 in advanced liver diseases, particularly in HCC and CC. This downregulation also correlated with poorer patient survival, suggesting that SUCLG1 may be a potential therapeutic target for age-associated liver cancers.

## 1. Introduction

The incidence of nearly all types of cancers increases dramatically with age [1]. Prolonged exposure to environmental carcinogens, lifestyle-related factors, and chronic inflammation cumulatively raise cancer risk [2]. Additionally, aging is associated with immunosenescence: the gradual decline in immune function that hampers the immune system’s ability to detect and clear precancerous and cancerous cells [3]. In fact, aging and cancer share several molecular mechanisms that directly connect the processes of cellular aging with cancer development, including genomic instability [4], telomere shortening [5], epigenetic alterations [6,7], cellular senescence [1], altered metabolism [8], and stem cell exhaustion [9]. Moreover, chronic, low-grade inflammation (‘inflammaging’) accumulates during aging, which disturbs system homeostasis and contributes to the development of pathogenic changes and age-related diseases such as cancer [10]. Recently, an inflammaging score-grading system has been suggested as a powerful method for prognosis assessment in cancer patients [11]. Furthermore, a longer lifespan increases cumulative exposure to environmental factors, such as radiation and pollutants, which contribute to DNA damage and, ultimately, cancer risk [12]. The connections between cancer and aging are thus both epidemiological and molecular. Understanding these connections could guide the development of therapies that target aging processes to reduce cancer risk and promote healthier aging.

Liver disease is typically age-associated, although there is some epidemiological controversy [13,14]. The global prevalence of liver disease is substantial and continues to rise due to multiple factors, including viral hepatitis, alcohol use, and metabolic disorders like non-alcoholic fatty liver disease (NAFLD). Liver diseases are responsible for approximately two million deaths annually, with one million attributed to complications of cirrhosis and another million to primary liver cancer, primarily hepatocellular carcinoma (HCC) [15]. HCC is the most common type of primary liver cancer, accounting for about 75–85% of all liver cancers. Cholangiocarcinoma (CC), a cancer of the bile duct, is less common than HCC and is subdivided into intrahepatic and extrahepatic cholangiocarcinoma. HCC and CC are both highly lethal liver cancers with significant mortality rates worldwide often due to late diagnoses and limited treatment options. HCC has a high mortality rate due to its aggressive nature and common late-stage diagnosis, with a 5-year survival rate of less than 20% in most countries [16]. Like HCC, CC has a poor prognosis, with an overall 5-year survival rate under 10% globally, mainly due to difficulties in early detection and the cancer’s resistance to chemotherapy and radiation [17].

How molecular drivers in HCC and CC, and in tumors in general, differ among patients of different ages is scarcely understood [13,18,19,20]. Recent studies comprehensively characterized transcriptomic alterations in relation to patients’ age across cancer types or multiple normal tissues, providing an omics view of age-associated alterations in health [21] and malignancy [22]. In this study, we extracted top gene candidates from a meta-analysis of 127 publicly available transcriptomic datasets from mice, rats, and humans, identifying a transcriptomic signature of aging across species and tissues (i.e., B2M, C1qA, and SUCLG1) [21], and tested their expression by qPCR and their correlation with disease progression in 48 tissue samples covering several liver disease stages (i.e., fatty liver, hepatitis, cirrhosis, HCC, and CC) and normal tissues, supported by bioinformatics analyses in The Cancer Genome Atlas (TCGA).

## 2. Materials and Methods

### 2.1. Human Tissue

Human normal cDNA array (Origene Technologies, catalogue number: HMRT504), human brain cancer cDNA array (Origene Technologies, catalogue number: HBRT502) and human liver cancer cDNA array (Origene Technologies, catalogue number: LVRT501) were used to assess the relative gene expression of B2M, C1qA, and SUCLG1 across brain and liver tissues. All available clinical information, including pathology reports and tissue images used in the microscopic examination, is recorded for each patient individually in the clinical information files provided by Origene Technologies; they can be found on the manufacturer’s website: https://www.origene.com/hbrt502-tissuescan-brain-cancer-cdna-array-i (for the brain array) and https://www.origene.com/lvrt501-tissuescan-liver-cancer-cdna-array-i (for the liver array).

### 2.2. Real-Time qPCR

TissueScan plates were removed from −20 °C storage and allowed to warm to room temperature for 20 min, followed by centrifugation at 1000× *g* for 1 min. To prepare the mixes for each of the genes, we used SG qPCR Master Mix (2×) (EurX, Gdansk, Poland; E0402-01, final concentration 1 × 2.5 mM MgCl2), ROX Solution (EurX, Gdansk, Poland; E0402-01, final concentration of 50 nM), forward and reverse primers (at final concentration of 300 nM), and nuclease-free water (EurX, Gdank, Poland; E0402-01). Since QuantStudio DX was used, we converted the arrays from 96-well to 384-well format. In short, 25 μL of the prepared gene-specific mix was added per well with a lyophilized cDNA sample. The plate was centrifuged for 1 min at 1000× *g*, vortexed, centrifuged a second time for 1 min at 1000× *g*, and allowed to stand at room temperature for 15 min to facilitate cDNA reconstitution. The plate was then vortexed and centrifuged for 1 min at 1000× *g*. Each sample was loaded in duplicates (10 μL per well) in a 384-well plate. The plate was then centrifuged for 1 min at 1000× *g*. The following primer sequences were used for the PCR amplification:
**Gene****Forward Primer****Reverse Primer****Annealing****Temperature (°C)**Actin βCTTCCTGGGCATGGAGTCCGCTCAGGAGGAGCAATGAT60GAPDHCATCACTGCCACCCAGAAGACTGATGCCAGTGAGCTTCCCGTTCAG60B2MGCCGCATTTGGATTGGATGAACCTAGAGCTACCTGTGGAGC58C1qAAAACATCAAGGACCAGCCGACGGTTCTTCCTGGTTGGTGA55SUCLG1CTTTTGCTGCTGCTGCCATTAAGCCTTGTCTTTTCCTGGCG58

### 2.3. Data Acquisition and Bioinformatics Analysis

The HCC and CC Illumina HiSeq-based gene expression datasets used in this study were downloaded from The Cancer Genome Atlas (TCGA) (https://www.cancer.gov/tcga), specifically from the GDC TCGA Liver Cancer (LIHC) and TCGA Bile Duct Cancer (CHOL) projects. The datasets comprised 377 HCC samples, 59 control HCC samples, 36 CC samples, and 9 control CC samples. Clinical information associated with both HCC and CC, specifically age at index (age of the patient at the diagnosis), vital status (survival status: ‘Alive’ or ‘Dead’), and days to death (time in days from diagnosis to death), was also retrieved from TCGA and integrated with the gene expression data using the Pandas Python library v1.5.3. Tumor samples were compared with controls for the assessment of the differential expression of the SUCLG1 gene. Tumor samples were analyzed independently of the controls to study the association between SUCLG1 expression profiles and patient survival. Gene expression data for the SUCLG1 gene were provided as FPKM values. Survival information was related to the overall survival of patients from the time of diagnosis to death, regardless of cause. To investigate the relationship between SUCLG1 expression and patient age in both HCC and CC, Spearman’s rank correlation analysis was performed using Python library SciPy v1.9.3. The analysis was conducted using SUCLG1 gene expression data and patient age, with both variables converted to numeric values, and missing or invalid data points were excluded. The statistical significance of the correlation was determined by the *p*-value, where a *p*-value less than 0.05 was considered statistically significant.

For the SUCLG1 gene, patients were divided into two groups (high expression and low expression) according to an identified optimal expression cut-point. This was achieved by exhaustively applying a log-rank test to multiple configurations of the two groups of patients. The optimal cut-point was defined as the SUCLG1 expression value that minimized the statistical significance of the log-rank test, thereby indicating the greatest separation in overall survival probability between the two groups. Kaplan–Meier survival curves were then generated using the KaplanMeierFitter function from the lifelines Python library v0.27.8 [23], which was then used to visualize the differences in survival probabilities over time. The statistical significance of survival differences between groups was assessed using the log-rank test, from which *p*-values and chi-square statistics were extracted. To compare the distributions of the HCC and CC groups for the gene SUCLG1, the Mann–Whitney U test was performed using the mannwhitneyu function of the Python library SciPy.

### 2.4. Statistical Analysis

Relative gene expression was calculated by the delta-delta Ct method. We used two housekeeping genes—actin beta (ACTB) and glyceraldehyde-3-phosphate dehydrogenase (GAPDH)—for normalization. The delta Ct for the housekeeping genes was calculated by taking the geometric mean of ACTB Ct and GAPDH Ct. Statistical analyses were conducted in GraphPad Prism (version 10.2.3). Multiple Mann–Whitney tests were performed to compare relative expression levels of B2M, C1qA, and SUCLG1 between the groups. To correct for multiple testing, we applied the Bonferroni–Dunn method. Asterisks indicate the level of significance: <0.05 (*), <0.01 (**), and <0.001 (***). Boxplots were created using GraphPad Prism (version 10.2.3).

## 3. Results

### 3.1. SUCLG1 Is Downregulated in Liver Disease and Cancers

We based our study on a global meta-analysis of aging that employed 127 microarray and RNA-Seq datasets from humans, mice, and rats and applied machine learning alongside enrichment methods [21]. That global meta-analysis across various tissues and species identified 449 genes overexpressed with age and 162 underexpressed with age [21]. We then selected three of the top upregulated or downregulated genes in aging for further studies in liver diseases/cancer: B2M (upregulated with age), C1qA (upregulated with age), and SUCLG1 (downregulated with age) (Table 1).

The selected genes exert quite distinct functions. B2M, or β2 microglobulin, is a component of MHC Class I molecules, found on the cell surface of all nucleated cells. C1qA encodes the A-chain polypeptide of serum complement subcomponent C1q, which associates with C1r and C1s to yield the first component of the serum complement system. SUCLG1 encodes the alpha subunit of the heterodimeric enzyme succinate coenzyme A ligase. This enzyme is targeted to the mitochondria and catalyzes the conversion of succinyl CoA and ADP or GDP to succinate and ATP or GTP. To study the role of these selected age-associated genes in the progression of liver diseases, we used a human liver cancer cDNA array obtained from liver biopsies of patients either healthy (*n* = 9) or with hepatitis (*n* = 3), fatty liver (*n* = 5), cirrhosis (*n* = 5), HCC (*n* = 24) and CC (*n* = 3). Demographic details and diagnosis of subjects are reported in Table 2. While B2M and C1qA did not display any change across the spectrum of liver disease, SUCGL1 mRNA expression decreases significantly and progressively with the deterioration of liver disease, reaching the lowest levels in malignant settings, i.e., HCC and CC (Figure 1). These findings seemed to be specific to hepatic malignancies, as none of the three genes (B2M, C1qA, and SUCGL1) displayed significant variations in their expression levels across a spectrum of human brain cancers [healthy (*n* = 3), meningioma (*n* = 25), astrocytoma (*n* = 8), oligoastrocytoma (*n* = 2), oligodendroglioma (*n* = 5), glioblastoma multiforme (*n* = 2), hemangiopericytoma (*n* = 2) (Appendix A)]. Demographic details and diagnoses of these subjects are reported in Appendix A. Furthermore, in our study cohort, the expression of SUCLG1 in liver tissues did not correlate with age (Appendix A).

### 3.2. Survival Analysis of SUCLG1 Expression in HCC and CC Patients

Next, we explored The Cancer Genome Atlas (TCGA), a landmark cancer genomics program that molecularly characterized over 20,000 primary cancers, and matched normal samples spanning 33 cancer types, including HCC and CC. The Mann–Whitney U test was performed to compare the distributions of the SUCLG1 gene between the HCC and CC groups and respective control samples. The analysis involved 377 HCC samples, 59 control HCC samples, 36 CC samples, and 9 control CC samples.

The analysis revealed a highly significant difference between the groups, with a *p*-value of 1.860560 x 10^-11^ and a U statistic of 11204.5, indicating a strong separation in the distribution of SUCLG1 expression between HCC or CC and their control groups, respectively (Figure 2). The Kaplan–Meier survival curves for the SUCLG1 gene expression levels in HCC and CC are shown in Figure 3 and Figure 4. Patients were stratified into two groups based on the optimal cut-point for SUCLG1 expression: high expression (blue) and low expression (orange). For CC, clinical information was available for all 36 samples, allowing for their inclusion in the Kaplan–Meier survival analysis. Conversely, of the 377 HCC samples, 354 had the necessary clinical information to generate the Kaplan–Meier survival curve.

For HCC, the log-rank test also indicated a significant survival difference between the high- and low-expression groups, with a *p*-value of 0.0419 and a chi-square value of 4.14. High SUCLG1 expression, i.e., ≥12.58, was associated with better survival outcomes than lower expression (<12.58) (Figure 3). Specifically, the high-expression group consisted of 24 samples, while the low-expression group included 330 samples. Similarly, for CC, the log-rank test revealed a significant difference in survival between the two groups, with a *p*-value of 0.0123 and a chi-square value of 6.26. Patients with high SUCLG1 expression, i.e., ≥10.39, exhibited better survival probabilities than those with lower expression (<10.39) (Figure 4). In this case, the high-expression group included 23 samples, while the low-expression group consisted of 13 samples.

Furthermore, our analysis of SUCLG1 expression levels and patient age in both the HCC and CC cohorts revealed differing trends. In the HCC cohort, the relationship between SUCLG1 expression and age was weak (Spearman correlation coefficient: 0.0596, *p*-value = 0.1131). Conversely, in the CC cohort, a significant negative correlation between age and SUCLG1 expression was observed (Spearman correlation coefficient: −0.4248, *p*-value = 0.0002), suggesting that older patients with lower SUCLG1 expression may have a poorer prognosis.

These results suggest that SUCLG1 expression is a significant predictor of survival in both HCC and CC patients, with higher expression correlating with improved survival outcomes.

## 4. Discussion

HCC and CC are a major challenge in oncology due to their aggressive nature and limited treatment options. Current treatments for HCC or CC, such as surgical resection, liver transplantation, and systemic therapies, often fail to provide long-term survival benefits. The identification of novel therapeutic targets is essential to improve patient outcomes. The shared mechanisms between aging and cancer suggest targeting aging processes may help mitigate cancer risk. Interventions like senolytics (agents that clear senescent cells), caloric restriction, and exercise have shown potential in reducing inflammation, enhancing DNA repair, and improving metabolic health, all of which may delay both aging and cancer development [1,2,3,4]. Cellular senescence is a hallmark in the progression of liver disease, HCC, and CC [24,25,26,27]. However, highly investigated senolytics such as dasatinib and quercetin may not be effective in preclinical models of HCC, and may even exacerbate the progression of age-associated liver disease [28,29]. Hence, new age-associated therapeutic targets for liver cancers should be identified. Here, we based our ex vivo and in silico study on a meta-analysis of 127 transcriptomic datasets from mice, rats, and humans, identifying a transcriptomic signature of aging across species and tissues [21], and identified SUCLG1 as a commonly downregulated protein in both HCC and CC tissue, which also associates to lower patient survival. SUCLG1 encodes an enzyme involved in the citric acid cycle, catalyzing the conversion of succinyl-CoA and ADP or GDP to succinate and ATP or GTP. This enzyme is essential for mitochondrial function and energy metabolism—processes that are often dysregulated in cancer. Mutations in the SUCLG1 gene have been linked to mitochondrial DNA depletion syndromes (MDDSs), characterized by severe metabolic dysfunctions [30]. Recently, Yan et al. showed that SUCLG1 restricts succinyl-CoA levels to suppress the succinylation of mitochondrial RNA polymerase (POLRMT), maintaining mtDNA transcription and mitochondrial biogenesis [31]. As the ‘Warburg effect’ is characterized by a metabolic shift in cancer cells from glucose catabolism via the citric acid cycle to anaerobic glycolysis, even in the presence of adequate oxygen levels, it is conceivable that SUCLG1 downregulation could be a possible candidate in the modulation of HCC and CC metabolic rewiring [32,33]. Metabolic changes are hallmarks of both aging and tumorigenesis. During aging, the build-up of metabolic byproducts and waste molecules in the circulation and within tissues negatively affects blood flow, oxygenation, and tissue stiffness. Altogether, these age-driven changes lead to metabolic reprogramming in different cell types, including liver parenchyma, which is particularly susceptible to stiffness and fibrosis [34]. The metabolic changes that occur during aging, including changes in SUCGL1 activity, might thus help create a favorable environment for liver tumorigenesis [35,36], although this notion has been debated regarding the very elderly [13,37]. Previous studies have linked SUCLG1 to various metabolic disorders and non-hepatic cancers, particularly leukemia, suggesting its potential as a therapeutic target [31,38,39,40]. Chen et al. showed that the transient overexpression of SUCLG1, PCK2, and GLDC can inhibit the progression of renal cancer cells [38], suggesting that a similar therapeutic approach might prove valuable also in HCC and/or CC preclinical models. As mentioned, SUCLG1 catalyzes the conversion of succinyl-CoA to succinate. Potential anti-cancer therapeutic strategies should target a boost in SUCLG1 activity and/or expression. In principle, SUCLG1 activity is enhanced by the direct supplementation of succinyl-CoA to patients, which, however does not seem a suitable strategy, since CoA esters are not membrane-permeable and therefore succinyl-CoA is unlikely to penetrate the cell membrane [41]. More promising approaches targeting SUCLG1 and other candidate genes, which are under investigation to treat MDDS, involve lentiviral or adeno-associated vector-mediated gene therapy [42]. In turn, the latter technologies, targeting genes such as adenovirus-thymidine kinase (ADV) or p53, are also currently being tested in clinical trials enrolling HCC patients [43]. However, it remains unpredictable whether gene therapy against SUCLG1 or other targets may impact overall survival or adverse events in aging HCC patients. Succinylation has been shown to be associated with worse patient survival prognosis in HCC. An altered expression of succinylation-related genes is linked to poor outcomes in HCC patients [44], supporting our findings. SUCGL1 is an integral part of the succinylation post-translational modification process, which affects various proteins, including histones [45]. The biological role of succinylation is relatively newly under investigation, but as the addition of a succinyl group introduces a relatively large structural moiety (100 Da) (bigger than acetylation (42 Da) or methylation (14 Da)), it is believed to significantly affect protein structure and function [46]. To the best of our knowledge, this is the first report associating SUCLG1 with CC and patient survival. However, our study does not provide a validation of SUCLG1 as a therapeutic target in preclinical or clinical models for HCC/CC or age-associated diseases, which remains to be explored in further studies. The identification of age-associated genes and pathways provides a deeper understanding of the molecular mechanisms underlying SUCLG1’s involvement in HCC and CC and highlights potential targets for therapeutic intervention.

## 5. Conclusions

Tissue biopsies with molecular phenotypes related to liver disease progression show downregulated mRNA levels of SUCLG1, most drastically observed in the malignant liver disease stages of HCC and CC. Low SUCLG1 expression levels among HCC and CC patients are also associated with poorer survival. The crucial role of SUCLG1 in metabolic homeostasis and its inhibition with age and in malignant liver disease stages present a rationale for further elucidating the ‘driving’ role of SUCLG1 in age-associated cancers.

## Figures and Tables

**Figure 1 cancers-17-00339-f001:**
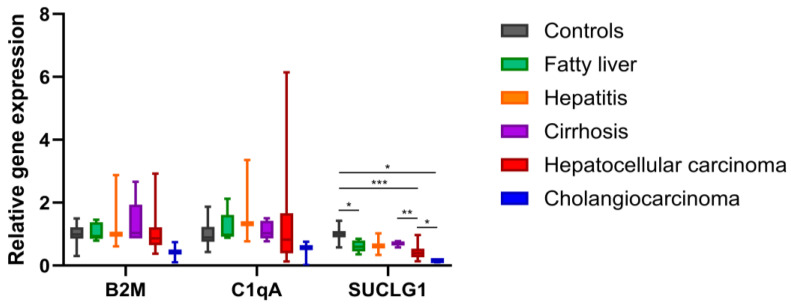
SUCLG1 mRNA levels decrease with liver-tissue deterioration alterations. Expression mRNA levels of B2M, C1qA, and SUCLG1 in tissues with normal appearance (*n* = 9) or tissue specimen with diagnosis: Fatty liver (*n* = 5), hepatitis (*n* = 3), cirrhosis (*n* = 5), hepatocellular carcinoma (*n* = 24) or cholangiocarcinoma (*n* = 3). Significant differences are indicated by asterisks: <0.05 (*), <0.01 (**), and <0.001 (***). All tissue samples in the liver cancer cohorts are biopsied from the tumor tissue area.

**Figure 2 cancers-17-00339-f002:**
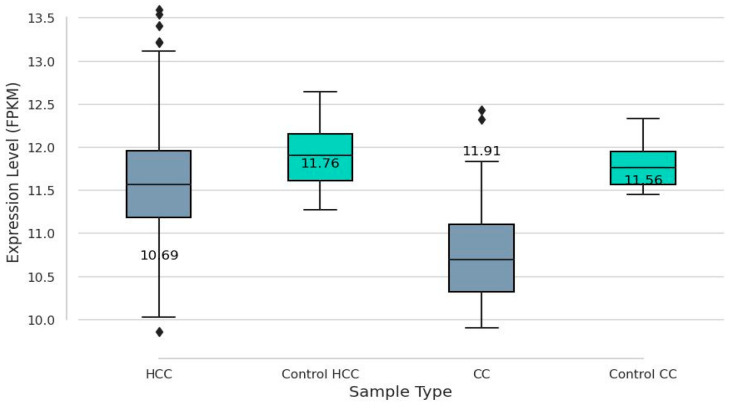
Boxplot of SUCLG1 expression levels in tumor and control samples. The plot compares SUCLG1 expression levels (FPKM) in hepatocellular carcinoma (HCC) and cholangiocarcinoma (CC) tumor samples with their respective normal control tissues. The analysis included 377 HCC tumor samples, 59 control HCC samples, 36 CC tumor samples, and 9 control CC samples. HCC tumor samples exhibit a median SUCLG1 expression of 10.69 FPKM, while the corresponding controls show a higher median expression of 11.76 FPKM. Similarly, CC tumor samples have a median SUCLG1 expression of 11.91 FPKM compared to a median of 11.56 FPKM in controls.

**Figure 3 cancers-17-00339-f003:**
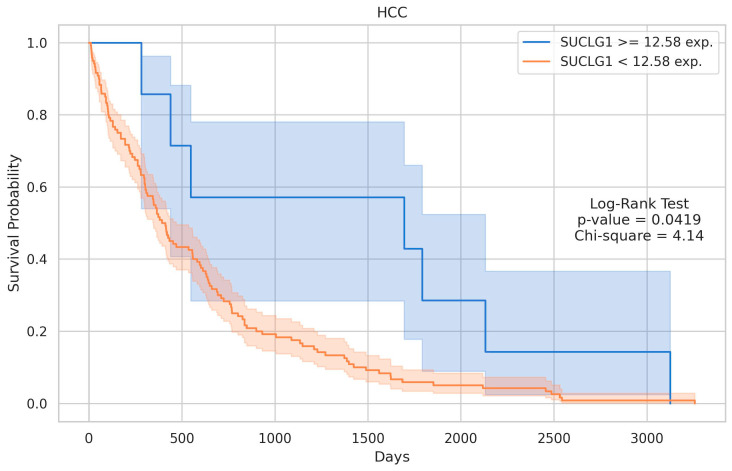
Kaplan–Meier survival curves for SUCLG1 expression in HCC. Kaplan–Meier survival plots display the survival probabilities for patients grouped by SUCLG1 expression levels in hepatocellular carcinoma (HCC). In HCC, high SUCLG1 expression (≥12.58) was associated with improved survival compared to low expression (<12.58) (log-rank test: *p* = 0.0419, χ2 = 4.14). The high-expression group comprised 24 samples, while the low-expression group included 330 samples.

**Figure 4 cancers-17-00339-f004:**
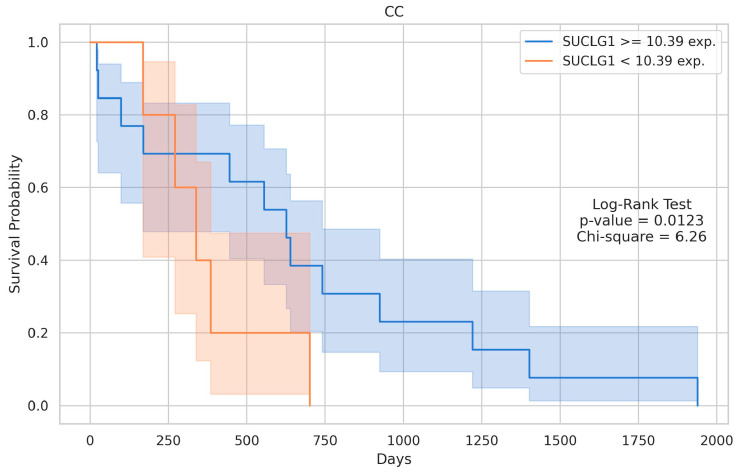
Kaplan–Meier survival curves for SUCLG1 expression in CC. Kaplan–Meier survival plots display the survival probabilities for patients grouped by SUCLG1 expression levels in cholangiocarcinoma (CC). In CC, patients with high SUCLG1 expression (≥10.39) had significantly better survival compared to those with low expression (<10.39) (log-rank test: *p* = 0.0123, χ2 = 6.26). The high-expression group included 23 samples, while the low-expression group consisted of 13 samples.

**Table 1 cancers-17-00339-t001:** Selection for genes differentially expressed with age across tissues and across species.

Gene	Gene Full Name	With Age	*p*-Value	Source (For a Complete List of Database See Appendix A of Original Study https://www.aging-us.com/article/202648/text)
C1qa	complement C1q A chain	Up	**3.54 × 10^−22^**	Microarray and RNA-Seq datasets
B2M	Beta-2-microglobulin	Up	**2.55 × 10^−20^**	Microarray and RNA-Seq datasets
SUCLG1	Succinate-CoA ligase GDP/ADP-forming subunit alpha	Down	**4.11 × 10^−9^**	Microarray and RNA-Seq datasets

**Table 2 cancers-17-00339-t002:** Characteristics of the liver cohort patients.

Patient n	Gender	Age	Sample Diagnosis	Patient Diagnosis	Tumor Grade	Stage
P1	Male	81	Normal	HCC	Not applicable	Not applicable
P2	Male	73	Normal	HCC	Not applicable	Not applicable
P3	Male	71	Normal	HCC	Not applicable	Not applicable
P4	Male	86	Normal	HCC	Not applicable	Not applicable
P5	Male	52	Normal	Granuloma	Not applicable	Not applicable
P6	Female	33	Normal	Hyperplasia	Not applicable	Not applicable
P7	Male	66	Normal	HCC	Not applicable	Not applicable
P8	Male	68	Normal	HCC	Not applicable	Not applicable
P9	Female	79	Hepatitis	HCC	Not applicable	Not applicable
P10	Male	68	Hepatitis	HCC	Not applicable	Not applicable
P11	Female	58	Hepatitis	HCC	Not applicable	Not applicable
P12	Male	73	Fatty liver	HCC	Not applicable	Not applicable
P13	Female	32	Fatty liver	Hyperplasia	Not applicable	Not applicable
P14	Male	79	Fatty liver	HCC	Not applicable	Not applicable
P15	Male	56	Fatty liver	HCC	Not applicable	Not applicable
P16	Male	26	Fatty liver	HCC	Not applicable	Not applicable
P17	Female	31	Adenoma	Adenoma	Not reported	Not reported
P18	Male	71	Cirrhosis	HCC	Not applicable	Not applicable
P19	Male	43	Cirrhosis	HCC	Not applicable	Not applicable
P20	Male	60	Cirrhosis	HCC	Not applicable	Not applicable
P21	Male	50	Cirrhosis	HCC	Not applicable	Not applicable
P22	Male	77	Cirrhosis	HCC	Not applicable	Not applicable
P23	Male	81	HCC	HCC	AJCC G1: Well-differentiated	I
P24	Male	79	HCC	HCC	AJCC G2: Moderately differentiated	I
P25	Female	61	HCC	HCC	AJCC G1: Well-differentiated	I
P26	Female	58	HCC	HCC	Not reported	I
P27	Male	66	HCC	HCC	AJCC G3: Poorly differentiated	I
P28	Female	63	HCC	HCC	AJCC G2: Moderately differentiated	II
P29	Male	73	HCC	HCC	AJCC G2: Moderately differentiated	II
P30	Male	68	HCC	HCC	Not Reported	II
P31	Male	60	HCC	HCC	AJCC G3: Poorly differentiated	II
P32	Female	62	HCC	HCC	AJCC G1: Well-differentiated	II
P33	Male	60	HCC	HCC	AJCC G2: Moderately differentiated	II
P34	Male	77	HCC	HCC	AJCC G1: Well-differentiated	II
P35	Male	63	HCC	HCC	AJCC G2: Moderately differentiated	II
P36	Female	39	HCC	HCC	AJCC G1: Well-differentiated	IIIA
P37	Male	43	HCC	HCC	AJCC G3: Poorly differentiated	IIIA
P38	Female	79	HCC	HCC	AJCC G2: Moderately differentiated	IIIA
P39	Male	56	HCC	HCC	AJCC G2: Moderately differentiated	IIIA
P40	Male	71	HCC	HCC	AJCC G2: Moderately differentiated	IIIA
P41	Male	86	HCC	HCC	AJCC G1: Well-differentiated	IIIA
P42	Male	26	HCC	HCC	AJCC G2: Moderately differentiated	IIIA
P43	Male	68	HCC	HCC	AJCC G2: Moderately differentiated	IIIA
P44	Male	21	HCC	HCC	AJCC G2: Moderately differentiated	IV
P45	Male	70	HCC	HCC	AJCC G3: Poorly differentiated	IV
P46	Female	62	CC	CC	AJCC G2: Moderately differentiated	I
P47	Female	78	CC	CC	AJCC G2: Moderately differentiated	I
P48	Male	66	CC	CC	Not reported	IV

## Data Availability

The original contributions presented in this study are included in the article. Further inquiries can be directed to the corresponding author. The raw data supporting the conclusions of this article will be made available by the authors on request.

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
