# Peer review of "Downregulation of Aging-Associated Gene SUCLG1 Marks the Aggressiveness of Liver Disease"

_cancers, 2025, doi:10.3390/cancers17030339_

Round 1
Reviewer 1 Report
Comments and Suggestions for Authors
The authors have demonstrated aging-associated gene SUCLG1 corelated with the prognosis of liver disease. The concept is interesting and the data are striking, especially shown in Fig. 4 and 5. The authors showed less SUCLG1 expressions are positively correlated with poor prognosis in HCC and CC. However, there are little data for the relationship between age and prognosis in the larger cohort. There are several concerns need to be addressed to make the paper more sophisticated.
-Reference numbers are not listed in the text, making it hard to understand what is known so far.
-In Fig. 1, please clarify whether liver biopsies were collected from tumor itself or non-tumor area (adjacent) especially in HCC or CC patients.
-Please describe the sample numbers in each group in each figure (1-5).
-Any correlation between SUCLG1 expressions and age in your cohort?
-Same as above, any correlation between SUCLG1 and age in the cohort shown in Fig. 4 or 5? Older patients showed inferior prognosis due to the less expression of SUCLG1?
-The results from human brain cancers should be listed in the supplementary data because these results are less relevant to the mainstream of the manuscript.
Author Response
The authors have demonstrated aging-associated gene SUCLG1 corelated with the prognosis of liver disease. The concept is interesting and the data are striking, especially shown in Fig. 4 and 5. The authors showed less SUCLG1 expressions are positively correlated with poor prognosis in HCC and CC. However, there are little data for the relationship between age and prognosis in the larger cohort. There are several concerns need to be addressed to make the paper more sophisticated.
We sincerely thank the Reviewer for his/her comments and efforts.
-Reference numbers are not listed in the text, making it hard to understand what is known so far.
We apologize for this mistake. We have now included the reference numbers within the text.
-In Fig. 1, please clarify whether liver biopsies were collected from tumor itself or non-tumor area (adjacent) especially in HCC or CC patients.
All samples are acquired from Origene Technologies in a ready-to-use qPCR array format (catalogue number: LVRT501). As per the information provided by Origene Technologies, the samples were selected based on the tumor content, which has been determined by microscopic analysis. The exact tumor content (presented as percentage) is available in the pathology reports, which can be found on the manufacturer’s website: https://www.origene.com/lvrt501-tissuescan-liver-cancer-cdna-array-i (for the liver array).
In Table 2 and Supplemental Table 1, we have indicated the tissue biopsy sample diagnosis, on which we have based the sample stratification into normal or diseased groups. In all Figures showing liver or brain cancer, all samples in the cancer groups are collected from the tumor area.
-Please describe the sample numbers in each group in each figure (1-5).
We have adjusted the Figures accordingly. We have now included the sample numbers for each group in the descriptions of Figures 1, 2, 3, and 4. Specifically, we have also described the sample numbers in the text associated with each figure.
-Any correlation between SUCLG1 expressions and age in your cohort?
We performed non-parametric Spearman correlation and did not find a significant correlation between SUCLG1 expression and age. We revised the manuscript to include the results from the addressed question in the supplementary data.
-Same as above, any correlation between SUCLG1 and age in the cohort shown in Fig. 4 or 5? Older patients showed inferior prognosis due to the less expression of SUCLG1?
Thank you for pointing this out. To address your questions, we performed a statistical analysis to examine the correlation between SUCLG1 expression levels and patient age in both cohorts (HCC and CC). Below are the results:
- HCC Cohort:
â—‹ Spearman correlation coefficient: 0.0596 (p-value = 0.1131)
The Spearman correlation does not support a significant relationship (p-value > 0.05), suggesting that the relationship between SUCLG1 expression and age in the HCC cohort is likely weak or negligible.
- CC Cohort:
â—‹ Spearman correlation coefficient: -0.4248 (p-value = 0.0002)
In the CC cohort, the Spearman analyses indicate a low negative correlation between SUCLG1 expression and age.
To further clarify, we analyzed the distribution of age in patients stratified by SUCLG1 expression levels (high vs. low) for both HCC and CC. The histograms (shown below) visualize the age distributions for the two expression groups.

Figure. Age distribution in Expression Groups: HCC (left) and CC (right)
Regarding prognosis, our results suggest that in CC, older patients may have inferior prognosis due to lower SUCLG1 expression. This hypothesis is supported by the observed significant negative correlation between age and SUCLG1 expression in CC. However, in HCC, the relationship between age and SUCLG1 expression does not appear to significantly impact prognosis.
These results have been incorporated into the Results (3.2. Survival Analysis of SUCLG1 Expression in HCC and CC Patients) and Methods (Data Acquisition and Bioinformatics Analysis) sections of the manuscript.
-The results from human brain cancers should be listed in the supplementary data because these results are less relevant to the mainstream of the manuscript.
We agree with the Reviewer. We have listed the results from the brain cancer analysis in the Supplementary data.
Reviewer 2 Report
Comments and Suggestions for Authors
Dear Author, I give you the following comment. Please address this in your manuscript to enhance the readability and understanding of your manuscript.
Major Comments:
- Methodology and Data Analysis
- Could the authors elaborate on how the top gene candidates (B2M, C1qA, SUCLG1) were selected from the pan-analyses? Were additional criteria or thresholds applied for their inclusion in this study?
- Sample Diversity
- The study analyzed 48 tissue samples covering different liver disease stages. Were there any limitations in the demographic or clinical diversity (e.g., age, sex, comorbidities) of these samples, and how might this affect the generalizability of the findings?
- Functional Validation
- Beyond the observed correlation of SUCLG1 expression with disease progression, did the authors perform any functional assays to confirm its direct role in liver disease aggressiveness?
- Comparison with Other Genes
- How does the downregulation of SUCLG1 compare in magnitude and significance to the other top candidates (B2M and C1qA)? Could this suggest a more pivotal role for SUCLG1 in disease progression?
- Therapeutic Potential
- The authors suggest that SUCLG1 might be a therapeutic target. Are there existing therapies or ongoing research targeting this gene or its pathway, and how might these findings align or diverge from those efforts?
Minor Comments:
- Terminology Clarification
- The term “inflamm-aging” is used in the introduction. Could a brief explanation or reference be provided for readers unfamiliar with this concept?
- Gene Expression Measurement
- What reference genes were used for normalization during the qPCR analysis, and were their expression levels consistent across the different liver disease stages?
- Figure Representation
- If figures or graphs are provided, could the authors ensure that the sample sizes for each liver disease stage are clearly indicated to aid in data interpretation?
- Statistical Analysis Details
- Could the authors specify the statistical tests used to evaluate the significance of SUCLG1 downregulation and its correlation with survival outcomes?
- Future Directions
- The discussion briefly mentions the therapeutic targeting of SUCLG1. Could the authors include a sentence on what specific experimental approaches or studies are planned to explore this avenue further?
These questions aim to address both overarching concerns and specific technical details that could impact the robustness and clarity of the study's findings.
Best Regards
Comments on the Quality of English LanguageFine
Reviewer 3 Report
Comments and Suggestions for Authors
Liver cancer is an important medical problem. In this regard, the study of the mechanisms of its development is an urgent task.
Comments:
1. The abstract should be structured in sections, including materials and methods, results, etc.
2. The materials and methods section needs to be improved. The study design as bioinformatics data analysis and clinical part should be described more clearly. It is necessary to describe in the materials and methods how patients were recruited, inclusion criteria, what examinations they underwent, etc.
3 Tables 2 and 3 need to be improved. Information on patients who underwent Real-time qPCR should be added. Clinical, laboratory data of patients, comorbidities, stages of disease medications taken are not clear.
4. Need to add a table with datasets (127 datasets) that were analyzed in the current study.
5. A table with identified differentially expressed genes (449+162) from section 3.1 needs to be added
6. Table 1 needs improvement. Why is the “statistics” column blank? The “Source” column is also not informative (which databases)?
7. The discussion section needs to be improved: discuss the results of other similar studies, discuss the prospects for future studies, discuss the limitations of the current study.
Round 2
Reviewer 1 Report
Comments and Suggestions for Authors
The authors have successfully addressed my concerns. The manuscript is well organized. I have no further comment.
Author Response
Thank you.
Reviewer 3 Report
Comments and Suggestions for Authors
The authors answered my questions and added additional data, which improved the quality of the article. However, I did not find any Supplementary files mentioned in the text.
Author Response
We apologise with the Reviewer for having forgotten to attach the file. We include it now.
